# PaTTA-ID: Practical Test-Time Adaptation for Person Re-Identification

## Abstract

Existing test-time adaptation (TTA) methods for person re-identification (re-ID) assume unrealistic scenarios: a large target gallery is available in advance, ignores temporal correlations in streaming input, and all identities are guaranteed to exist in the gallery set. Furthermore, they rely on server-side settings where data from multiple cameras are aggregated in advance, which is unrealistic for edge device applications on a single-camera. Therefore, they experience performance degradation in practical real-world deployments due to domain gaps between the training (source) data and the unseen (target) gallery streams. In this work, we introduce a practical scenario of test-time adaptation for person re-ID tailored for online streaming environment on resource-constrained edge devices, where a small predefined query set is registered in advance and unlabeled large gallery data continuously arrive from a single camera stream. We propose a novel framework to address this practical problem, called PaTTA-ID, that enables effective adaptation from two complementary perspectives. First we devise *Input Distribution Compensation*, which employs query-guided sampling and contrastive adaptation to compensate the bias of streaming inputs and promote cross-camera discriminability. Moreover, we investigate *Model Drift Compensation*, which prevents the bias toward the current camera stream via camera invariant learning and query feature compensation. Experimental results evaluated on four benchmark datasets compared with nine baselines demonstrate that the proposed PaTTA-ID achieves state-of-the-art performance surpassing existing TTA methods.

## 1 Introduction

Person re-identification (re-ID) is the task of matching a query image of a target individual against gallery images captured from different cameras. It has drawn much attention due to its practical applicability to real-world scenarios such as surveillance, augmented reality, and video analytics. Recent re-ID methods Ye et al. (2021); Chen et al. (2023); Yuan et al. (2025) have substantially improved the performance of the model when the test data comes from the same target dataset. However, in real-world scenarios, diverse datasets are generated across different locations and at different times. This makes existing approaches suffer from a domain gap caused by distribution shifts such as camera styles and environmental changes.

Recent studies such as BNTA (Han et al., 2022) and TEMP (Adachi et al., 2024) have explored test-time adaptation (TTA) for person re-identification by adapting a pretrained model to an unseen target domain. However, both assume that the entire gallery, consisting of images captured by multiple cameras in the target domain, is available beforehand. This assumption is unrealistic in practice, where gallery instances continually arrive over time. In addition, they assume that every observed person belongs to a predefined query set, which overlooks non-query individuals who frequently appear in real-world environments. These methods also ignore the strong temporal correlation in person appearances naturally arising in sequential streams. These limitations underscore the need for a more practical TTA framework that works online and handles incomplete, temporally ordered, and open-world gallery streams.

In this paper, we propose PaTTA-ID, a practical test-time adaptation framework for person re-identification, motivated by real-world deployment scenarios. Unlike prior settings, we assume a predefined query set representing target individuals of interest, such as suspects or missing persons.

This assumption is practical, as query identities are typically known in advance and can be registered before deployment. Our framework continuously adapts the model to an unlabeled stream of gallery data captured from a single camera, where non-query persons are present and person appearances exhibit strong temporal correlation. This practical setting introduces two major challenges: (i) *Noisy and Correlated Data Streams*, where gallery inputs are temporally correlated and dominated by non-query persons, injecting noise and redundancy into adaptation and hindering the learning of camera-invariant representations; and (ii) *Model Drift*, where continual updates on gallery data from a single camera stream bias the embedding space and cause the pre-extracted query features to become misaligned with the evolving feature space.

To address these challenges, we propose PaTTA-ID, which enables adaptation from two complementary perspectives: (i) *Input Distribution Compensation*, achieved through query-guided sampling that retains only reliable gallery instances and contrastive adaptation that strengthens cross-camera discriminability. Together, these mechanisms compensate for model drift and biased input distributions, enabling robust adaptation; and (ii) *Model Drift Compensation*, achieved through camera-aware sampling based camera-invariant learning together with query drift compensation that updates stored query features using estimated drift vectors. We evaluate PaTTA-ID with four person re-ID benchmarks of Market1501, CUHK03, MSMT17, and LPW under our practical scenario. We compare PaTTA-ID with nine state-of-the-art TTA methods (Nado et al., 2020; Wang et al., 2020; Gong et al., 2022; Wang et al., 2022; Niu et al., 2023; Yuan et al., 2023; Gong et al., 2023; Han et al., 2022; Adachi et al., 2024), including recent studies that address TTA in person re-ID (Han et al., 2022; Adachi et al., 2024). Our evaluation with multiple person re-ID benchmarks demonstrates that PaTTA-ID outperforms existing methods. For instance, PaTTA-ID achieved 50.1% Rank-1 accuracy on the most challenging dataset (CUHK03), outperforming the best baseline Wang et al. (2020) by 22.7% and the recent re-ID work Adachi et al. (2024) by 37.2%.

Our main contributions are summarized as follows: (i) *Practical TTA Setting for Person re-ID*, a realistic test-time adaptation setting for person re-ID that considers real-world deployments, which has not been investigated in the field. (ii) We propose a PaTTA-ID framework that enables effective adaptation through two complementary strategies that compensate for the input distribution and the model drift. (iii) The experimental results with multiple person re-ID benchmarks demonstrate that PaTTA-ID outperforms existing methods under practical TTA setting.

## 2 RELATED WORK

**Person Re-Identification.** Person re-ID aims to retrieve query images of a given identity across different cameras. Supervised learning methods (Sun et al., 2018; Luo et al., 2019; Ye et al., 2021) and unsupervised learning methods (Lin et al., 2019; Dai et al., 2022; Cho et al., 2022; Chen et al., 2021) with pseudo-labeling or clustering techniques have been actively explored to address this task. Recently, transformer-based methods (He et al., 2021; Zhang et al., 2023) and self-supervised pretraining (Fu et al., 2021; 2022; Chen et al., 2023) on large-scale unlabeled datasets have significantly advanced the state of the art. In addition, post-processing techniques (Song et al., 2025) have been proposed to remove the camera bias problem when the model encounters an unseen domain. Domain generalization methods (Liao & Shao, 2022; Ni et al., 2023; Dou et al., 2023) have also been introduced to improve robustness against unseen target domains. However, prior re-ID studies have been developed and evaluated under static settings, focusing on benchmark-style analysis where both training and inference are performed offline with pre-collected datasets.

**Test-Time Adaptation.** Test-time adaptation (TTA) aims to adapt a source-pretrained model to an unlabeled test data stream. A widely studied setting is online TTA, where each mini-batch of target data is used once to update model parameters and then discarded. Representative approaches include entropy minimization on test predictions (Wang et al., 2020), adjusting normalization statistics to handle distribution shifts (Gong et al., 2022), and using a teacher model with weight restoration (Wang et al., 2022). More recent methods address challenges in realistic streams, such as using statistics and reweighting old samples (Yuan et al., 2023), filtering high-entropy samples from mixed domains (Niu et al., 2023), or reducing prediction sharpness to cope with noisy data (Gong et al., 2023).

**Test-Time Adaptation for Person Re-Identification.** Recently, test-time adaptation methods for person re-ID, BNTA (Han et al., 2022) and TEMP (Adachi et al., 2024), have been proposed to

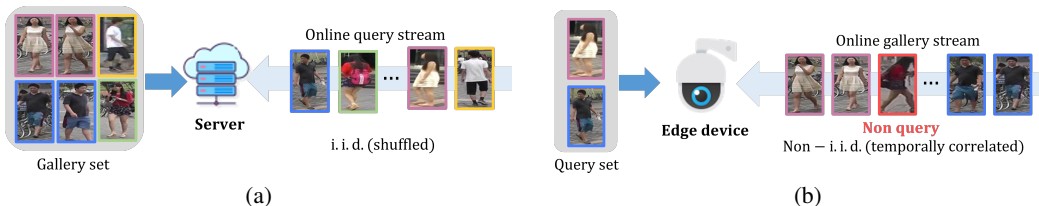

Figure 1: Comparison of TTA scenarios for person re-ID. (a) Previous and (b) the proposed scenarios.

adapt pretrained re-ID models to the target domain. Both methods assume that the entire gallery set of the target domain is available in advance, which might be unrealistic in practice. BNTA (Han et al., 2022) employs a part classification loss on a subset of gallery images to guide the model in learning part-aware representations, and then performs offline inference with the query and gallery images. TEMP (Adachi et al., 2024), in contrast, performs online adaptation by minimizing the prediction uncertainty of sequentially arriving queries, where uncertainty is estimated via cosine similarity between query features and pre-extracted gallery features. However, these assumptions are impractical for real-world deployments: they require centralized access to the entire target gallery in advance and neglect the streaming nature of surveillance systems, where gallery data arrives sequentially, non-query persons are dominant, and storage or privacy constraints often prevent maintaining a complete gallery set. We elaborate on these limitations in Section 3.1.

## 3 METHODOLOGY

### 3.1 PROBLEM DEFINITION AND CHALLENGES

Test-time adaptation (TTA) enables models pretrained on a source domain to cope with distribution shifts in unseen target domains using only unlabeled test data, which is particularly important in person re-ID, where domain gaps—such as camera style and environmental changes—cause severe performance degradation. Prior TTA methods for person re-ID (Fig. 1(a)) rely on several assumptions: (i) a large gallery set of the target domain is available in advance and stored in a centralized server, (ii) query images arrive sequentially in random order from multiple cameras, and (iii) the identity of each query is guaranteed to exist in the gallery set. These assumptions, however, are rarely satisfied in practice. In real-world scenarios: (i) pre-collecting and storing a large gallery is often impractical on edge devices, as person images are continually collected over time, (ii) persons instead arrive sequentially with strong temporal correlations, and (iii) many observed individuals do not belong to the given query set, i.e., they are *non-query* persons. To address these limitations, we propose a practical TTA scenario for person re-ID, motivated by real-world deployment on edge devices (Fig. 1(b)). In real applications, target individuals of interest (e.g., suspects or missing children) are typically known in advance, so we assume a predefined query set. A source-pretrained model is deployed to a camera, which continuously receives unlabeled gallery streams containing both query and non-query persons with temporal correlations. Then person re-ID is performed from the query set to the online gallery streams.

Considering this practical TTA scenario for person re-ID, two major challenges arise. First, from the input perspective, streaming gallery data pose challenges in two ways: (i) they are temporally correlated and lack cross-camera diversity, since they originate from only one camera at a time; and (ii) many *non-query* identities outside the predefined query set are continuously introduced. Such skewed and noisy input distributions hinder stable adaptation and gradually erode the discriminability of learned representations. Second, continual parameter updates induce *model drift*, which appears in two forms: (i) the embedding space becomes biased toward the current camera stream—adapting solely to a single camera style risks overfitting and undermines the camera-invariance required for cross-camera retrieval; and (ii) the fixed query feature memory becomes stale and misaligned with the evolving embedding space.

### 3.2 METHODOLOGY OVERVIEW

To address the aforementioned challenges, we propose Practical Test-Time Adaptation for person re-ID (PaTTA-ID), as illustrated in Fig. 2. PaTTA-ID enables effective adaptation from two com-

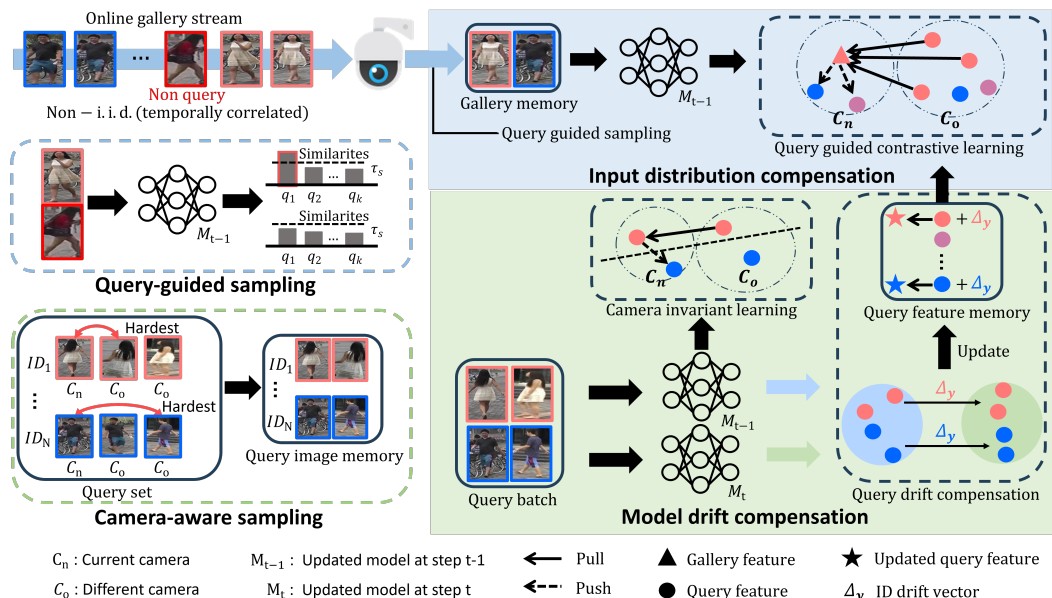

Figure 2: Overview of PaTTA-ID.

plementary perspectives: (i) *Input Distribution Compensation* which leverages query features as anchors to filter noisy gallery data and enhance cross-camera discriminability; and (ii) *Model Drift Compensation* which enhances camera-invariant and query-specific representations for the current camera stream and mitigates query feature drift.

Before gallery persons arrive, PaTTA-ID first initializes two memories: the query feature memory $\mathcal{F}$ and the query image memory $\mathcal{Q}$. The source-pretrained model $\mathcal{M}_s$ is used to extract features of all query instances and store them in $\mathcal{F}$, while a small subset of query images, selected by a camera-specific sampling strategy, is stored in $\mathcal{Q}$. As gallery persons sequentially arrive from the online stream, inference is performed in an online manner: given a gallery instance $g_i$ with its feature $f_i$ extracted by $\mathcal{M}^{t-1}$, we compute the cosine similarity between $f_i$ and all query features in $\mathcal{F}$ to retrieve its identity. After inference, query-guided sampling strategy is adopted to select high-confident, query-relevant samples and store them in the gallery memory $\mathcal{G}$ (see Appendix A for details). Once a sufficient number of gallery samples have been accumulated, the model is updated at the $t$-th step, where the previous model $\mathcal{M}^{t-1}$ is optimized into $\mathcal{M}^t$ using gallery samples and a randomly drawn query batch from $\mathcal{Q}$. For model adaptation, query guided contrastive learning and camera-invariant learning is performed by using the following losses: $\mathcal{L}_{\text{total}} = \mathcal{L}_{\text{qca}} + \mathcal{L}_{\text{ce}} + \mathcal{L}_{\text{tri}}$. Furthermore, PaTTA-ID compensates for query feature drift in $\mathcal{F}$ by estimating identity-wise drift vectors from queries in $\mathcal{Q}$ and applying them to update the memory.

### 3.3 INPUT DISTRIBUTION COMPENSATION

A key challenge in streaming-based person re-ID lies in the biased input distribution: a gallery stream is temporally correlated, dominated by non-query instances, and restricted to a single camera style at a time. Such skewed distributions of multiple gallery streams from different cameras hinder the adaptation by causing over-representation of certain identities and amplifying the noise. Under these conditions, the model should still enhance its discriminability for gallery persons to enable reliable cross-camera retrieval, despite being continuously exposed to biased and noisy inputs.

To compensate for the biased input distributions, we adopt a Query-Guided Sampling scheme that leverages the query features as anchors to retain only confident gallery instances in the gallery memory $\mathcal{G}$. To employ independent and identically distributed (i.i.d.) batches, we prevent the memory from being dominated by a few IDs, where the IDs whose samples are excessively stored are replaced first (see Appendix A for details). Then, we exploit the query feature memory $\mathcal{F}$ together with the samples stored in $\mathcal{G}$ to guide the model adaptation. Specifically, let $x_{g,i}^{t-1}$ denote the feature of a

gallery sample in $\mathcal{G}$ extracted by $\mathcal{M}^{t-1}$, and let $l_{g,i}$ be its pseudo label. The positive and negative query sets of the features are defined as

$$\mathcal{P}_i = \{z_j \mid l_{f,j} = l_{g,i}\}, \quad \mathcal{N}_i = \{z_j \mid l_{f,j} \neq l_{g,i}\}, \tag{1}$$

where $l_{f,j}$ denotes the ground-truth label of the $j$-th feature $z_j$ in $\mathcal{F}$. To pull $x_{g,i}^{t-1}$ closer to the samples in the positive set and push it away from challenging negatives, we compute $\rho_i$ for $x_{g,i}^{t-1}$ given by

$$\rho_i = \frac{1}{|\mathcal{P}_i|} \sum_{z \in \mathcal{P}_i} \log \frac{\exp(s(x_{g,i}^{t-1}, z)/\tau_t)}{\sum_{y \in \mathcal{P}_i \cup \mathcal{D}_i} \exp(s(x_{g,i}^{t-1}, y)/\tau_t)}, \tag{2}$$

where $\mathcal{D}_i$ is the set of the negative samples in $\mathcal{N}_i$ having the top-k similarities to $x_{g,i}^{t-1}$, and $s(\cdot, \cdot)$ is the cosine similarity. Finally, we define the query-guided contrast adaptation loss as

$$\mathcal{L}_{\text{qca}} = -\frac{1}{|\mathcal{G}|} \sum_{i \in I(\mathcal{G})} \rho_i, \tag{3}$$

where $I(\cdot)$ means the index set. By combining the query-guided memory sampling with the proposed contrastive adaptation loss, PaTTA-ID compensates for the biased input distributions in streaming re-ID, alleviating the dominance of the non-query instances and the temporal correlation while enhancing discriminability across cameras.

### 3.4 MODEL DRIFT COMPENSATION

Online updates on a single-camera stream induce model drift, where the embedding space gradually shifts toward the streaming camera distribution, and leading the fixed query features in $\mathcal{F}$ gradually become misaligned with the updated features. As the person re-ID is a cross-camera retrieval task, it is essential to prevent such model drift problem. We address these issues with two complementary approaches as follows.

**Camera-Invariant Learning.** We employ a small number of query samples as auxiliary training data. For each identity in the query set, we store two representative queries in the query image memory $\mathcal{Q}$ via camera-aware sampling, which is shown in Figure 2. Specifically, for each identity, we first include one query image that was captured from the same camera with the current stream $c_n$. Then, we select another query image captured from a different camera from $c_n$, that yields the lowest similarity to the stored one. This strategy ensures that $\mathcal{Q}$ maintains both intra-camera and cross-camera diversity while keeping the memory cost low.

To conduct cross-camera invariant learning, we use the cross-entropy loss and the triplet loss for model training. Specifically, at every update step $t$, we randomly sample a mini-batch from $\mathcal{Q}$ to generate the query batch. Let $q_i$ denote the $i$-th sample in the query batch, $l_{q,i}$ its identity label, and $x_{q,i}^{t-1}$ the feature of $q_i$ extracted by $\mathcal{M}^{t-1}$. The classifier predicts the probability $p(l_{q,i} \mid x_{q,i}^{t-1})$ for each sample that the sample has the ground truth identity label, and the cross-entropy loss is computed as

$$\mathcal{L}_{\text{ce}} = -\frac{1}{B} \sum_{i=1}^{B} \log p(l_{q,i} \mid x_{q,i}^{t-1}), \tag{4}$$

where $B$ is the number of samples in the query batch. Also, for each anchor $x_{q,i}^{t-1}$ in the batch, we apply the triplet loss as follows:

$$\mathcal{L}_{\text{tri}} = \frac{1}{B} \sum_{i=1}^{B} \left[ \max_{x_p:l_p=l_{q,i}} \|x_{q,i}^{t-1} - x_p\|_2 - \min_{x_n:l_n \neq l_{q,i}} \|x_{q,i}^{t-1} - x_n\|_2 + \alpha \right]_+, \tag{5}$$

where $x_p$ and $x_n$ are the hardeset positive and hardeset negative samples, respectively, with respect to the anchor $x_{q,i}^{t-1}$, and $\alpha$ is the margin hyperparameter. By continuously optimizing both losses with the query batches from $\mathcal{Q}$, the model learns to preserve the cross-camera invariance and enhances the discriminability of query features, thereby mitigating the model drift.

**Query Feature Compensation.** As model parameters are updated during the online adaptation, the query features initially stored in $\mathcal{F}$ gradually become misaligned with the evolving feature space, leading to a drift of the query feature. Figure 3 illustrates this phenomenon. Here, *Old Query* refer to the query features extracted by the source model $\mathcal{M}_s$, representing the initial features stored in $\mathcal{F}$. *Oracle Query* and *Gallery* are obtained by extracting the same query and gallery images using the fully updated model trained on the entire gallery stream, respectively. We compared the similarity distributions between the query features (Old or Oracle) and their positive gallery features (same identity but captured from different cameras). As shown, Old Queries (red) yield substantially

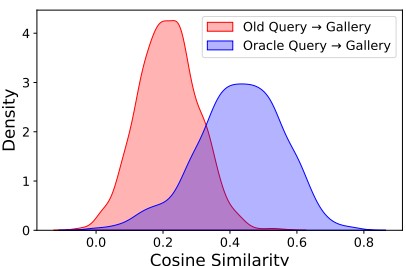

Figure 3: Similarity distribution illustrating the query drift problem.

lower similarity with their positives, while Oracle Queries (blue) maintain higher similarity. In such cases, the fixed features in $\mathcal{F}$ become incompatible with the continuously updated feature space during the gallery stream adaptation. We may re-extract the features of all the query instances after each update of the model, however, this requires the storage space for the query images as well as the high computation to repeatedly extract the features.

For each sample $q_i$ in a mini-batch sampled from $\mathcal{Q}$, we extract the features by using both $\mathcal{M}^t$ and $\mathcal{M}^{t-1}$, denoted as $x_{q,i}^t$ and $x_{q,i}^{t-1}$, respectively. Then we estimate identity-wise feature drift as

$$\Delta_l = \frac{1}{|B_l|} \sum_{i \in B_l} \left( x_{q,i}^t - x_{q,i}^{t-1} \right), \tag{6}$$

where $B_l$ denotes the set of the query samples with the label $l$. The drift vector $\Delta_l$ captures the average shift in the feature space for the identity $l$ associated with two consecutive update steps of the model. With this identity-specific correction, each query feature $z_j$ in $\mathcal{F}$ associated with the label $l$ is updated as

$$z_j \leftarrow z_j + \Delta_l, \quad \forall j \text{ s.t. } l_{f,j} = l. \tag{7}$$

This simple yet effective rule continuously aligns the query features in $\mathcal{F}$ with the evolving feature spaces, thereby mitigating *model drift*.

## 4 EXPERIMENTAL RESULTS

### 4.1 EXPERIMENAL SETUP

**Datasets.** We adopted four widely-used person re-ID datasets, Market1501 (Zheng et al., 2015), CUHK03-NP (Zhong et al., 2017), LPW (Xu et al., 2025), and MSMT17 (Wei et al., 2018) to conduct our TTA experiments. Table 1 shows the statistics of the datasets. We used one dataset as a source to pretrain the model, and the remaining datasets are individually set as the target datasets.

To construct our practical TTA setting, we first define the query set of the target dataset by uniformly sampling 20 % of the total query identities. This design choice reflects real-world scenarios, where only a limited number of query persons are available in advance, while the majority of observed individuals in the gallery are non-query images. Using all identities as queries would be unrealistic, as it assumes an impractically exhaustive query pool. Furthermore, uniform sampling ensures that query identities appear consistently across the entire gallery stream, rather than being concentrated in only the early portion. For the gallery stream, we split the gallery set according to camera labels and generate one continuous stream per camera, simulating an online deployment scenario.

**Evaluation Metrics.** In our practical re-ID TTA setting, evaluation is performed online for each incoming gallery instance. For gallery instances that have the same identities as the persons in the predefined query set, we use the widely used mean averaged precision (mAP) and cumulative matching characteristics (CMC) Rank1 score to measure retrieval performance. In addition, as our work is the first to explicitly define *non-query* persons in re-ID, we introduce a new evaluation protocol, termed Non-Query Aware Receiver Operating Characteristic (NQ-ROC), to assess the ability of a model to reject them. Specifically, we use the top-1 cosine similarity between a gallery

Table 1: Statistics of person re-ID datasets used in our experiments.

| Dataset | # identities | # images | # query | # cameras |
|---|---|---|---|---|
| Market-1501 (Zheng et al., 2015) | 1,501 | 32,668 | 3,368 | 6 |
| CUHK03-NP (Zhong et al., 2017) | 1,467 | 14,097 | 1,400 | 2 |
| LPW (Xu et al., 2025) | 1,751 | 25,177 | 2,456 | 4 |
| MSMT17 (Wei et al., 2018) | 4,101 | 126,441 | 11,659 | 15 |

Table 2: Comparison with state-of-the-art methods on the target datasets of Market1501, CUHK03-NP, and LPW. Backbone models are trained on the source dataset of MSMT17. Best scores are boldfaced.

| Backbone | Methods | Market1501 | | | CUHK03-NP | | | LPW | | |
|---|---|---|---|---|---|---|---|---|---|---|
| | | mAP | Rank1 | NQ-ROC | mAP | Rank1 | NQ-ROC | mAP | Rank1 | NQ-ROC |
| Strong Re-ID | No adapt | 49.6 | 65.0 | 74.4 | 25.2 | 16.7 | 57.6 | 51.1 | 53.6 | 71.6 |
| | BN Stats (Nado et al., 2020) | 55.3 | 69.7 | 71.9 | 37.6 | 26.0 | 60.5 | 55.5 | 60.3 | 71.9 |
| | TENT (Wang et al., 2020) | 53.6 | 67.3 | 69.2 | 38.9 | 27.4 | 60.4 | 56.4 | 59.8 | 70.7 |
| | NOTE (Gong et al., 2022) | 52.6 | 66.6 | 68.0 | 37.1 | 25.6 | 59.8 | 56.4 | 59.8 | 68.6 |
| | CoTTA (Wang et al., 2022) | 16.2 | 18.4 | 53.3 | 8.1 | 3.5 | 52.2 | 13.0 | 11.3 | 51.0 |
| | SAR (Niu et al., 2023) | 55.2 | 69.7 | 71.8 | 37.7 | 26.2 | 60.5 | 55.6 | 60.4 | 71.9 |
| | RoTTA (Yuan et al., 2023) | 55.7 | 70.8 | 68.2 | 36.7 | 25.8 | 59.4 | 57.2 | 61.7 | 70.5 |
| | SoTTA (Gong et al., 2023) | 56.0 | 72.4 | 69.6 | 38.2 | 27.3 | 56.7 | 55.0 | 59.0 | 69.1 |
| | BNTA (Han et al., 2022) | 41.8 | 57.9 | 66.0 | 22.4 | 12.5 | 55.1 | 42.7 | 43.7 | 63.4 |
| | TEMP (Adachi et al., 2024) | 44.1 | 59.6 | 71.5 | 21.1 | 12.9 | 55.8 | 46.3 | 47.9 | 68.3 |
| | PaTTA-ID | **76.0** | **85.5** | **81.3** | **61.5** | **50.1** | **66.7** | **69.0** | **71.4** | **75.6** |
| CLIP Re-ID | No adapt | 50.6 | 67.6 | 73.1 | 38.3 | 26.5 | 63.4 | 57.9 | 62.4 | 71.3 |
| | BN Stats (Nado et al., 2020) | 49.2 | 60.6 | 66.0 | 38.1 | 26.2 | 60.6 | 56.4 | 60.2 | 69.4 |
| | TENT (Wang et al., 2020) | 49.1 | 59.5 | 65.6 | 38.5 | 26.7 | 60.8 | 56.8 | 60.1 | 69.2 |
| | NOTE (Gong et al., 2022) | 47.4 | 57.8 | 65.9 | 38.8 | 26.6 | 61.0 | 57.7 | 62.4 | 68.5 |
| | CoTTA (Wang et al., 2022) | 25.2 | 29.2 | 52.1 | 14.1 | 6.9 | 51.4 | 24.5 | 24.1 | 53.2 |
| | SAR (Niu et al., 2023) | 49.2 | 60.5 | 66.0 | 38.2 | 26.3 | 60.6 | 56.4 | 60.1 | 69.4 |
| | RoTTA (Yuan et al., 2023) | 50.3 | 63.2 | 64.5 | 40.0 | 28.0 | 60.5 | 58.3 | 61.5 | 69.6 |
| | SoTTA (Gong et al., 2023) | 51.0 | 65.5 | 66.1 | 40.7 | 27.9 | 60.6 | 58.9 | 63.4 | 70.7 |
| | BNTA (Han et al., 2022) | 48.8 | 65.4 | 71.5 | 35.7 | 23.3 | 61.1 | 58.7 | 61.0 | 72.0 |
| | TEMP (Adachi et al., 2024) | 47.5 | 63.7 | 71.2 | 33.9 | 22.3 | 61.9 | 54.9 | 57.8 | 69.0 |
| | PaTTA-ID | **73.3** | **85.1** | **79.6** | **66.8** | **55.7** | **70.0** | **73.3** | **77.3** | **77.8** |

feature and the query feature memory as the decision score, and compute the area under the resulting ROC curve. NQ-ROC summarizes the trade-off between correctly rejecting non-query instances and mistakenly matching them to queries across varying decision thresholds. Note that we measure mAP, Rank-1, and NQ-ROC per camera of the target domain, and report the average across all cameras.

**Implementation Details.** We adopted two backbones, Strong Re-ID (Luo et al., 2019) and CLIP Re-ID (Li et al., 2023), to train the model with the source dataset. For CLIP Re-ID, we used a ResNet50 (He et al., 2016) based network. Following their original papers and official codes, we trained the models for 120 epoch on the source dataset, and used Adam optimizer with initial learning rate of 0.00035, which is decayed by the value of 0.1 at epoch 40 and 70. For test-time adaptation, we used a test batch size of 64, and set the memory size of $\mathcal{G}$ to 64. Also, the model was updated every time 64 gallery persons arrived. We updated all weights of the model by using the Adam optimizer with a fixed learning rate of $0.00035$ and a weight decay of $0.0$. $\tau_t$ and $\alpha$ is empirically set to 1.0 and 0.3, respectively. We used Pytorch and a single NVIDIA RTX 3090 GPU.

## 4.2 COMPARISON WITH STATE-OF-THE-ART METHODS

We compare PaTTA-ID with state-of-the-art TTA baselines of TENT(Wang et al., 2020), NOTE(Gong et al., 2022), CoTTA(Wang et al., 2022), SAR(Niu et al., 2023), RoTTA(Yuan et al., 2023), SoTTA(Gong et al., 2023) and recent person re-ID test-time adaptation baselines of BNTA(Han et al., 2022) and TEMP(Adachi et al., 2024). **No adapt** evaluates the source model directly on the target gallery data without any adaptation. **BN Stats** updates the BN statistics of the source model with the target gallery data. For classification TTA methods, it is hard to directly adopt these methods

Table 3: Comparison with state-of-the-art methods on the target datasets of MSMT17, CUHK03-NP, and LPW. Backbone models are trained on the source dataset of Market1501. Best scores are boldfaced.

| Backbone | Methods | MSMT17 | | | CUHK03-NP | | | LPW | | |
| --- | --- | --- | --- | --- | --- | --- | --- | --- | --- | --- |
| | | mAP | Rank1 | NQ-ROC | mAP | Rank1 | NQ-ROC | mAP | Rank1 | NQ-ROC |
| Strong Re-ID | No adapt | 11.4 | 18.0 | 56.4 | 11.3 | 5.5 | 52.2 | 35.6 | 38.4 | 71.8 |
| | BN Stats (Nado et al., 2020) | 12.4 | 17.4 | 55.7 | 24.2 | 15.0 | 55.7 | 53.1 | 58.1 | 73.8 |
| | TENT (Wang et al., 2020) | 10.9 | 14.4 | 54.5 | 26.2 | 16.4 | 56.2 | 54.3 | 58.9 | 74.4 |
| | NOTE (Gong et al., 2022) | 10.2 | 13.7 | 54.6 | 24.4 | 15.1 | 52.2 | 51.3 | 54.3 | 68.9 |
| | CoTTA (Wang et al., 2022) | 1.9 | 1.9 | 51.7 | 4.0 | 1.0 | 49.6 | 8.3 | 6.8 | 51.0 |
| | SAR (Niu et al., 2023) | 12.5 | 17.4 | 55.7 | 24.3 | 15.1 | 55.7 | 53.1 | 58.1 | 73.9 |
| | RoTTA (Yuan et al., 2023) | 12.1 | 17.5 | 56.1 | 22.5 | 14.2 | 53.2 | 54.8 | 59.5 | 73.2 |
| | SoTTA (Gong et al., 2023) | 10.3 | 14.7 | 55.6 | 18.1 | 10.1 | 52.6 | 52.1 | 56.4 | 71.2 |
| | BNTA (Han et al., 2022) | 8.8 | 13.7 | 53.7 | 13.4 | 7.3 | 53.5 | 29.7 | 31.0 | 65.8 |
| | TEMP (Adachi et al., 2024) | 7.0 | 10.9 | 53.8 | 7.1 | 2.8 | 50.9 | 33.0 | 36.2 | 70.7 |
| | PaTTA-ID | **20.4** | **25.4** | **60.0** | **58.9** | **48.4** | **67.4** | **68.4** | **67.6** | **75.8** |
| CLIP Re-ID | No adapt | 12.3 | 18.5 | 57.9 | 25.7 | 16.2 | 58.2 | 48.5 | 51.3 | 76.7 |
| | BN Stats (Nado et al., 2020) | 10.2 | 13.8 | 54.2 | 29.5 | 19.3 | 56.1 | 55.5 | 59.8 | 74.5 |
| | TENT (Wang et al., 2020) | 10.3 | 13.6 | 54.1 | 31.1 | 20.9 | 56.5 | 56.2 | 60.5 | 74.8 |
| | NOTE (Gong et al., 2022) | 8.6 | 11.4 | 53.5 | 28.3 | 18.4 | 53.9 | 51.5 | 55.0 | 70.1 |
| | CoTTA (Wang et al., 2022) | 2.5 | 2.3 | 52.2 | 6.1 | 2.0 | 50.0 | 13.3 | 11.6 | 50.9 |
| | SAR (Niu et al., 2023) | 10.2 | 13.8 | 54.2 | 29.5 | 19.3 | 56.1 | 55.5 | 59.8 | 74.5 |
| | RoTTA (Yuan et al., 2023) | 10.2 | 14.0 | 55.6 | 29.6 | 19.6 | 55.8 | 57.6 | 61.6 | 76.3 |
| | SoTTA (Gong et al., 2023) | 10.3 | 14.5 | 55.5 | 32.0 | 23.2 | 56.1 | 55.3 | 58.8 | 74.7 |
| | BNTA (Han et al., 2022) | 11.0 | 16.9 | 56.3 | 21.5 | 12.4 | 55.0 | 43.3 | 45.0 | 74.3 |
| | TEMP (Adachi et al., 2024) | 8.1 | 12.3 | 55.2 | 23.6 | 14.4 | 55.6 | 48.8 | 53.1 | 75.7 |
| | PaTTA-ID | **18.3** | **23.9** | **58.9** | **64.2** | **54.4** | **70.8** | **72.7** | **74.4** | **79.4** |

since the identities in the target domain differ from those of the source domain. In an alternative way, we employ last classification layer that was trained on the source domain to compute logits and classification scores of target gallery persons, and apply the methods in their papers. For person re-ID TTA baselines, note that since the scenarios of their methods are different from ours, we fit their methods to our problem setting. BNTA(Han et al., 2022) assumes all gallery persons of the target domain are prepared in advance, and applies self-supervised losses to a small subset of gallery persons for adaptation. Differently, we apply the losses to the sequentially arriving gallery persons for adaptation since our problem setting assumes gallery persons come in sequentially. For TEMP(Adachi et al., 2024), we apply re-ID entropy loss to the sequentially arriving gallery persons, through estimating the cosine similarity between gallery features and pre-extracted query features.

Table 2 and Table 3 show the online evaluation results on various target domains. Existing methods yield some improvements on certain datasets but fail to provide sufficient gains due to model drift and the presence of noisy inputs. Similarly, person re-ID TTA methods such as BNTA and TEMP exhibit severe performance degradation, confirming that existing approaches cannot effectively adapt the model in our scenario, as they are not designed to handle temporally correlated streams with non-query persons. In contrast, PaTTA-ID achieves the best performance under our practical setting, demonstrating its ability to mitigate model drift and remain robust to noisy input streams.

## 4.3 ABLATION STUDY

**Effect of Each Module in PaTTA-ID.** In Table 4, we present an ablation study to evaluate the effectiveness of the individual components of PaTTA-ID. Here, Re-ID Entropy denotes the entropy loss proposed in TEMP (Adachi et al., 2024), where the loss is applied to samples in $\mathcal{G}$ for a fair comparison. QCA indicates the query-guided contrastive adaptation loss, QRL denotes camera-invariant learning that trains the model by using both $\mathcal{L}_{ce}$ and $\mathcal{L}_{tri}$, with query batch samples, and QFC corresponds to query feature compensation. As shown in the table, QCA alone leads to performance degradation, confirming that training only with gallery data from a single camera suffers from drift caused by distribution shift. In contrast, combining QRL and QFC with QCA yields huge improvements, demonstrating that leveraging queries through camera-aware sampling effectively regularizes the model and mitigates the drift of query features. These results validate that addressing biased input distributions and model drift provides a synergistic strategy for practical online TTA for

Table 4: Ablation study showing the effectiveness of each component in PaTTA-ID.

| Methods | Market1501 | | | CUHK03-NP | | | LPW | | |
|---|---|---|---|---|---|---|---|---|---|
| | mAP | Rank1 | NQ-ROC | mAP | Rank1 | NQ-ROC | mAP | Rank1 | NQ-ROC |
| No adapt | 49.6 | 65.0 | 74.4 | 25.2 | 16.7 | 57.6 | 51.1 | 53.6 | 71.6 |
| Re-ID Entropy | 23.9 | 23.8 | 57.0 | 14.8 | 6.1 | 53.5 | 26.5 | 23.0 | 56.3 |
| QCA | 30.4 | 32.5 | 56.9 | 23.6 | 13.9 | 54.2 | 32.4 | 29.3 | 60.0 |
| QRL | 66.7 | 80.6 | 76.9 | 47.7 | 36.3 | 61.9 | 60.6 | 64.1 | 71.4 |
| QRL + QCA | 69.0 | 82.0 | 77.0 | 49.8 | 36.6 | 61.5 | 65.4 | 69.2 | 73.9 |
| QRL + QFC | 73.0 | 83.3 | 80.2 | 57.0 | 45.5 | **66.9** | 66.0 | 68.7 | 74.2 |
| QRL + QFC + Re-ID Entropy | 55.1 | 58.3 | 68.1 | 41.2 | 24.9 | 59.2 | 51.4 | 46.7 | 64.6 |
| QRL + QFC + QCA | **76.0** | **85.5** | **81.3** | **61.5** | **50.1** | 66.7 | **69.0** | **71.4** | **75.6** |

person re-ID, improving the baseline performance by +26.8%, +23.9%, and +6.7% in averaged mAP, Rank-1, and NQ-ROC, respectively.

**Effect of camera-aware query sampling strategy.** We analyzed the effect of camera-aware sampling strategy in constructing query image memory $\mathcal{Q}$. Table 5 shows the results in Market1501, where all the methods perform sampling on each identity that are in the predefined query set. The results validate that our camera-aware sampling strategy, which samples the query image tailored to the camera stream, as well as selecting the hardest cross-camera query image provides the best performance.

Table 5: Comparison of query image memory sampling methods

| Method | mAP | Rank1 | NQ-ROC |
|---|---|---|---|
| Randomly sample 2 | 63.9 | 77.3 | 75.1 |
| Randomly sample 4 | 65.6 | 77.7 | 76.7 |
| 2 Cross-cam | 61.1 | 72.6 | 74.2 |
| 1 stream-cam + 1 cross-cam (easiest) | 71.7 | 83.8 | 81.2 |
| **1 stream-cam + 1 cross-cam (hardest)** | **76.0** | **85.5** | **81.3** |

**Impact of query ID ratios.** Figure 4 shows the varying performance in Market1501 when we set different ratios of queries to construct the query set. The results validate that our approach consistently outperforms the no-adapt baseline across all query ratios, demonstrating its robustness in adapting to the target domain. In particular, even when the predefined query set is very large (e.g., 90% of the total query identities in Market1501), our approach still yields clear improvements, highlighting its effectiveness under such conditions.

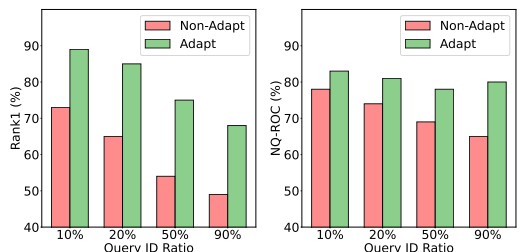

Figure 4: Performance comparison when using different query ratios to construct the query set.

## 5 CONCLUSION

In this paper, we introduced a Practical Test-Time Adaptation for Person Re-ID (PaTTA-ID), a framework designed for realistic deployment on edge devices where gallery data arrive sequentially from a single camera stream. Unlike prior test-time adaptation approaches that rely on centralized access to complete gallery sets, PaTTA-ID addresses the challenges of online person re-ID, including biased input distributions and model drift. Extensive experiments across multiple benchmarks demonstrated that PaTTA-ID consistently outperforms existing re-ID baselines and test-time adaptation methods, validating its effectiveness and practicality. We expect this pioneering work would encourage further research for robust re-ID systems in real-world streaming environments.

**Limitations and future work.** A limitation of our current setting is that it assumes a predefined query set, where each query is provided with multiple instances captured from diverse cameras, which may not always be feasible in fully open-world scenarios. As a future direction, we plan to investigate more challenging few-shot settings, where only a handful of query instances are available. Also, we will explore more advanced scenarios in which new query identities are incrementally introduced during deployment.

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

## A  ALGORITHM DETAILS

**Gallery memory sampling**   As aforementioned, gallery stream possess high correlation between test batches and includes huge amount of negative samples(non-query) that has different identity to query persons. To overcome this issue, we use query as an anchor to guide robust sampling and mimic independent and identically distributed(i.i.d) batch from temporally correlated data streams. The overall algorithm is illustrated in Algoritm 1.

---

**Algorithm 1:** Query-guided sampling

---

**Require:** Gallery stream $\{g_i\}_{i=1}^{K}$, gallery memory memory $\mathcal{G}$ with capacity $N$, threshold $\tau_s$
1: **for** each gallery instance **do**
2:    Extract feature $f_i$ from incoming $g_i$
3:    Compute confidence score $s_i = \max_{z \in \mathcal{Q}} s(z, f_i)$
4:    Assign pseudo-label $l_{g,i} = \arg\max_{z \in \mathcal{Q}} s(z, f_i)$
5:    **if** $s_i > \tau_s$ **then**
6:      **if** $|M| < N$ **then**
7:        Add $(g_i, l_{g,i}, s_i, a_i)$ to $\mathcal{G}$
8:      **else**
9:        $\mathcal{Y}^* \leftarrow$ ID(s) with the largest occupancy in $\mathcal{G}$
10:       **if** $l_{g,i} \notin \mathcal{Y}^*$ **then**
11:         Select ID $\hat{y} \in \mathcal{Y}^*$ having the **highest age**
12:         Remove the sample in ID $\hat{y}$ having the **lowest confidence score**
13:       **else**
14:         Remove the sample from ID $l_{g,i}$ having the **lowest confidence score**
15:       **end if**
16:       Add $(g_i, l_{g,i}, s_i, a_i)$ to $\mathcal{G}$
17:      **end if**
18:    **end if**
19: **end for**

---

Specifically, given a feature $f_i$ extracted from the incoming gallery data $g_i$ by using $\mathcal{M}^{t-1}$ and the query feature memory $\mathcal{F} = \{z_1, z_2, \ldots, z_{|\mathcal{F}|}\}$, we compute the cosine similarity between $f_i$ and each query feature $z_i$, and define the top-1 similarity score as the confidence score of gallery data $g_i$ as

$$s_i = \max_{q \in \mathcal{F}} \ s(z, f_i), \tag{8}$$

We then assign the pseudo label of $g_i$ as the label of the query instance as

$$l_{g,i} = \arg\max_{z \in \mathcal{F}} \ s(z, f_i). \tag{9}$$

In addition, we define the age of $g_i$ as $a_i$, where $a_i$ represents the number of times the sample $g_i$ has been used to update the model, and it is incremented by $+1$ each time $g_i$ participates in an update. To reduce the effect of non-query samples for adaptation, we store the gallery data if its predicted confidence score is higher than the predifined threshold $\tau_s$. When $\mathcal{G}$ reaches its capacity, we first identify the most prevalent IDs. Then we select the oldest identity(ID) based on the age of instances, and discard the instance with the lowest confidence score from the selected ID. In this way, we can maintain balances among IDs and reduce the impact of non-query samples for adaptation. With the stored samples in $\mathcal{G}$, we optimize $\mathcal{M}^{t-1}$ by using query-guided contrastive loss.

## B  ADDITIONAL EXPERIMENTS

**Advanced backbone**   In this section, we show additional comparative results of PaTTA-ID. Table 6 shows the results when we adopt SOLIDER Re-ID (Chen et al., 2023) as backbone model. We adopted SWIN-Tiny network which is pretrained on large-scale unlabeled dataset LuP (Fu et al., 2021), and then trained the model on the source dataset MSMT17 to generate backbone source model. As shown in the table, our PaTTA-ID consistently outperforms previous approaches across

Table 6: Comparison with state-of-the-art methods on target dataset : Market1501, CUHK03-NP , and LPW. Backbone models are trained on the source dataset: MSMT17. Best scores are boldfaced.

| Backbone | Methods | Market1501 | | | CUHK03-NP | | | LPW | | |
|---|---|---|---|---|---|---|---|---|---|---|
| | | mAP | Rank1 | NQ-ROC | mAP | Rank1 | NQ-ROC | mAP | Rank1 | NQ-ROC |
| SOLIDER Re-ID | BN Stats (Nado et al., 2020) | 71.4 | 83.4 | 82.9 | 59.7 | 47.5 | 70.1 | 74.4 | 78.4 | 77.6 |
| | TENT (Wang et al., 2020) | 71.7 | 83.6 | 82.9 | 60.2 | 47.9 | 70.1 | 74.7 | 78.7 | 77.7 |
| | NOTE (Gong et al., 2022) | 71.9 | 83.9 | 83.0 | 60.2 | 48.1 | 70.1 | 74.7 | 78.6 | 77.7 |
| | CoTTA (Wang et al., 2022) | 63.5 | 75.4 | 68.4 | 53.0 | 40.9 | 59.7 | 65.0 | 67.6 | 66.2 |
| | SAR (Niu et al., 2023) | 71.4 | 83.4 | 82.9 | 59.7 | 47.5 | 70.1 | 74.3 | 78.4 | 77.6 |
| | RoTTA (Yuan et al., 2023) | 71.4 | 83.4 | 82.9 | 59.7 | 47.5 | 70.1 | 74.4 | 78.4 | 77.6 |
| | SoTTA (Gong et al., 2023) | 71.5 | 83.4 | 82.9 | 59.9 | 47.6 | 70.1 | 74.6 | 78.6 | 77.6 |
| | TEMP (Adachi et al., 2024) | 71.3 | 83.2 | 82.8 | 59.6 | 47.3 | 70.1 | 74.4 | 78.4 | 77.6 |
| | PaTTA-ID | **79.2** | **89.2** | **86.6** | **73.1** | **62.3** | **75.3** | **78.4** | **81.3** | **80.2** |

Table 7: Comparison with state-of-the-art methods on target dataset : PRW, CDPS. Backbone models are trained on the source dataset: MSMT17. Best scores are boldfaced.

| Backbone | Methods | PRW | | | CDPS | | |
|---|---|---|---|---|---|---|---|
| | | mAP | Rank1 | NQ-ROC | mAP | Rank1 | NQ-ROC |
| Strong Re-ID | BN Stats (Nado et al., 2020) | 50.3 | 65.4 | 74.0 | 38.6 | 42.0 | 63.5 |
| | TENT (Wang et al., 2020) | 48.4 | 62.1 | 72.8 | 28.2 | 28.9 | 71.2 |
| | NOTE (Gong et al., 2022) | 49.2 | 63.7 | 73.8 | 32.1 | 31.7 | 60.1 |
| | CoTTA (Wang et al., 2022) | 13.8 | 14.6 | 53.4 | 7.5 | 6.2 | 49.7 |
| | SAR (Niu et al., 2023) | 50.3 | 65.4 | 74.0 | 38.7 | 42.1 | 63.3 |
| | RoTTA (Yuan et al., 2023) | 52.2 | 68.6 | 72.8 | 43.5 | 46.8 | **65.9** |
| | SoTTA (Gong et al., 2023) | 52.2 | 66.4 | 71.4 | 39.2 | 41.3 | 61.8 |
| | BNTA (Han et al., 2022) | 36.1 | 49.7 | 66.7 | 21.4 | 21.2 | 57.3 |
| | TEMP (Adachi et al., 2024) | 36.1 | 47.0 | 68.7 | 20.6 | 21.0 | 58.3 |
| | PaTTA-ID | **74.7** | **82.2** | **80.6** | **52.8** | **52.2** | 55.5 |

all benchmarks by a huge margin. Interestingly, most of baseline methods does not suffer from performance drop as the source model is well generalized due to large scale pretraining datasets. Our method aids from this generalized knowledge and achieves superior performance.

**Different target datasets** Moreover, we additionally use two large-scale challenging person search benchmark datasets of PRW Zheng et al. (2017) and CDPS Zhang et al. (2024) as target datasets to conduct more comparative experiments, We cropped the images of test dataset in PRW and CDPS by using the ground-truth(GT) bounding boxes and conducted TTA experiments. As shown in Table 7, PaTTA-ID outperforms all baseline methods in PRW dataset in all metrics by a huge margin. In CDPS, PaTTA-ID outperforms all baseline methods in terms of mAP and Rank1, demonstrating the effectivness of PaTTA-ID under our practical scenario.

## C ADDITIONAL ABLATION STUDY

**Effect of query-guided sampling strategy.** We analyzed the effect of query-guided sampling strategy in constructing gallery image memory $\mathcal{G}$. Table 8 shows the results in Market1501. FIFO retains the most recent samples by discarding the oldest ones as new data arrive. Time-Uniform (Reservoir) selects samples with equal probability over time, ensuring unbiased coverage of the

Table 8: Comparison of gallery image memory sampling methods

| Method | mAP | Rank1 | NQ-ROC |
|---|---|---|---|
| FIFO | 74.2 | 82.9 | 80.0 |
| Time uniform | 73.7 | 83.1 | 80.4 |
| **Query-guided sampling** | **76.0** | **85.5** | **81.3** |

entire stream. All the methods perform sampling on the sequentially arriving gallery person. The results validate that our query-guided sampling strategy, which samples the gallery person with high similarity with respect to query features in $\mathcal{F}$, provided the best performance.

## D    EXPERIMENT DETAILS

In the experiments, we use the official implementation of the baseline methods. Below, we detail the descriptions of the adopted hyperparameters and implementation specifics.

**TENT**    We set the learning rate as 0.00035 in all target datasets, which is same value as ours. We referred to their original codes [1] for implementations.

**COTTA**    CoTTA involves three hyperparameters: the augmentation confidence threshold $p_{th}$, the restoration factor $p$, and the exponential moving average (EMA) factor $m$. For consistency, we follow the settings recommended by the original authors, using $p = 0.01$ and $\alpha = 0.999$ in our implementation. We referred to their original codes [2] for implementations.

**SAR**    SAR is designed to adapt to varying batch sizes, and we used a batch size of 64 for fair comparison. We set the learning rate to 0.00035, the sharpness threshold to $\rho = 0.5$, and the entropy threshold to $E_0 = 0.4 \times \ln|Y|$, where $|Y|$ denotes the number of classes. In addition, we froze the top layer (layer4 for ResNet50) as in the original implementation, which SoTTA also adopts. We referred to their original codes [3] for implementations.

**NOTE**    We set the batch size to 64 for fair comparison. The soft-shrinkage width value is set to 4, and momentum value is set to 0.1. We used adam optimizer with a learning rate of 0.00035. We referred to their original codes [4] for implementations.

**ROTTA**    We set adam optimizer with a learning rate of 0.00035 and $\beta = 0.9$. We followed the hyperparameter settings reported by the authors, including the BN-statistic EMA rate $\alpha = 0.05$, the teacher model EMA rate $\nu = 0.001$, and the timeliness parameter $\lambda_t = 1.0$. We set the memory capacity to 64 for fair comparison. We referred to their original codes [5] for implementations.

**SOTTA**    We set the HUS memory size as 64 for fair comparison. The confidence threshold $C_o$ to 0.3, ADAM optimizer with a momentum value of 0.1, and learning rate 0.00035 is used. We set the value of entropy-sharpmess L2-norm constraint $\rho$ to 0.5. We referred to their original codes [6] for implementations.

**BNTA**    We set number of stripes to 6, and the margin to 0.3. We used adam optimizer with learning rate of 0.00035. Self-supervised SSL losses of their papers are applied to sequentially arriving gallery persons. We do not sample the gallery set using part nearest neighbor mathcing algorithm for the model adaptation as it violates our practical scenarios. We referred to the codes in [7] for implementations.

**TEMP**    We used a batch size of 64 for fair comparison. We used adam optimizer with learning rate of 0.00035, and weight decay of 0.0. $\tau$ is set to 0.0001. For re-ID entropy loss, the temperature value and k is set to 1.0 and 50, respectively, following their original codes. We referred to their original codes [8] for implementations.

## E    ADDITIONAL DISCUSSIONS

**Broader Impacts.**    Our work focuses on practical test-time adaptation for person re-identification, targeting realistic deployment scenarios on edge devices with single-camera streams. By addressing

---

[1] https://github.com/DequanWang/tent
[2] https://github.com/qinenergy/cotta
[3] https://github.com/mr-eggplant/SAR
[4] https://github.com/TaesikGong/NOTE
[5] https://github.com/BIT-DA/RoTTA
[6] https://github.com/taeckyung/SoTTA
[7] https://github.com/kzkadc/reid-tta
[8] https://github.com/kzkadc/reid-tta

biased input distributions and model drift in an online and resource-efficient manner, PaTTA-ID facilitates more robust and accessible deployment of re-ID systems in applications such as public safety, traffic monitoring, and video analytics. However, as with any advancement in re-identification technology, there is potential for negative societal impacts: it could also be misused for intrusive surveillance or violations of personal privacy. We therefore emphasize the importance of ethical guidelines and regulatory oversight in applying our method to real-world systems.

**Reproducibility.** The codes for reproducing the experiments will be provided.

