# OpenReview forum: "PaTTA-ID : Practical Test-Time Adaptation for Person Re-Identification"
_ICLR.cc/2026/Conference — ICLR 2026 Conference Withdrawn Submission_

### Official Review · Reviewer_7Hr9 · 2025-10-30

**Soundness:** 3
**Presentation:** 3
**Contribution:** 2
**Rating:** 4
**Confidence:** 4

**Summary:**

The topic of this paper is person re-identification for video surveillance. In this paper a new task of the broader ReID problem is introduced. A new practical scenario for re-ID, targeted for edge devices, where a small predefined query set is registered in advance, and model is adapted to specific camera. For this scenario a new framework is proposed. The experimental evaluation is organized as following. The model is trained on one dataset, and then applied to another dataset with adaptation. This is similar in nature to testing of the generalizable person re-ID methods, but with adaptation to the target dataset. Experimental evaluation demonstrates that the proposed method allows significant improvement in accuracy compared to other test-time adaptation re-ID methods and generalizable re-ID methods without adaptation.

**Strengths:**

1) New practical scenario of person re-ID with adaptation to target camera, which can be potentially applied in industrial systems with good results
2) Extensive evaluation on most of modern benchmarks (some of the results are in supplementary)
3) Extensive description of all model parameters and meta-parameters, which will allow re-implementation even if source-codes are not released

**Weaknesses:**

1) The are a number of details missing in the description and discussion of the proposed scenario. The overal goal is on-line adaptation to the targeted camera on edge device. However, all experiments have been performed using 3090 gpu, which is definetely not an edge device. What are the computational and memory requirements of the proposed model for inference and adaptation? How long the adaptation is taking time? What are the memory requirements for features, gallery images, etc? All this information is required to thoroughly assess whether the proposed method is suitable for the proposed scenario.

2) One important weakness is identified by the authors themselfs. Method requires a pre-defined query set with multiple instances captures from diverse cameras. In practice only 1 query image is provided per person and new queries are provided.

**Questions:**

1) How the method will perform if pre-defined query set has only 1 image per person, which is how it is in practice?
2) How the method will perform if new query images are provided?
3) Please, address the computational and memory requirements for the method inference and adaptation.

---

### Official Review · Reviewer_efJe · 2025-10-30

**Soundness:** 2
**Presentation:** 3
**Contribution:** 2
**Rating:** 4
**Confidence:** 4

**Summary:**

The paper proposes a realistic test-time adaptation setting for person re-ID tailored for online streaming environments and proposes a PaTTA-ID framework that enables effective adaptation through two complementary strategies that compensate for the input distribution and the model drift. The experimental results with multiple person re-ID benchmarks demonstrate that PaTTA-ID outperforms existing methods under the practical TTA setting.

**Strengths:**

- The paper proposes a new test-time adaptation setting for person re-ID tailored for online streaming environments.

- The experimental results show the effectiveness of the proposed method.

**Weaknesses:**

Please refer to Questions. Some parts of the proposed method are not clearly introduced.

**Questions:**

•	The memory Q stores only two representative queries. Given this limited size, Is the positive sample for each anchor uniquely determined, and how is the hardest positive sample selected? A clearer explanation of the sampling strategy is needed.

•	The test-time adaptation (TTA) setting in the prior work TEMP employs a fixed gallery and processes streaming query, whereas the proposed method uses a fixed query set and processes streaming gallery. The two settings share conceptual similarities—with some minor differences such as sampling 20% of the query set—it would be insightful to explore whether TEMP's setting can be adapted by using streaming query data from a single camera. A discussion on the interchangeability of these two settings would strengthen the analysis.

•	The design of Input Distribution Compensation bears a strong resemblance to the approach in TEMP, which also computes cosine similarity between the query feature and each gallery feature and selects the top-k most similar samples for further processing. Is the primary distinction merely that TEMP optimizes based on entropy minimization, while the proposed method employs a contrastive loss? If so, a more explicit comparison of the two objectives and their respective effects would be valuable.

•	In Figure 4, the performance degrades as the query set size increases, which is counterintuitive. One would expect that a larger query set provides more diverse and informative data, potentially improving model adaptation and performance. The authors should provide a justification for this unexpected trend.

•	The “No adapt” baseline results should be included in Tables 6 and 7 for completeness. This would allow for a more comprehensive comparison and better illustrate the actual gain brought by the proposed adaptation strategy across different settings.

---

### Official Review · Reviewer_5NWu · 2025-10-31

**Soundness:** 3
**Presentation:** 3
**Contribution:** 3
**Rating:** 6
**Confidence:** 4

**Summary:**

Existing test-time adaptation (TTA) methods for person re-identification (re-ID) rely on unrealistic assumptions (e.g., pre-available full target gallery, ignoring temporal correlation in streams), leading to poor performance on resource-constrained edge devices with single-camera streams; to address this, the paper proposes PaTTA-ID, a framework that combines Input Distribution Compensation (query-guided sampling + contrastive adaptation) and Model Drift Compensation (camera-invariant learning + query feature compensation) to tackle biased stream inputs and model drift. Experiments on 4 benchmarks (Market1501, CUHK03, MSMT17, LPW) show PaTTA-ID outperforms 9 baselines—e.g., 50.1% Rank-1 on CUHK03, 22.7% higher than the best baseline. Its contributions include: a practical streaming TTA setting for re-ID, the dual-compensation PaTTA-ID framework, and the new NQ-ROC metric for non-query rejection evaluation.

**Strengths:**

1. The proposed "edge-device single-camera streaming re-ID TTA" setting breaks the unrealistic "pre-available full gallery" assumption of existing methods, creating a new direction for deployment-oriented re-ID research. The NQ-ROC metric, the first to evaluate non-query rejection ability, complements the re-ID TTA evaluation system.
2. Experiments are systematic—covering 4 mainstream benchmarks, comparing 9 baselines (general TTA + re-ID-specific TTA), and using multi-metrics (mAP, Rank-1, NQ-ROC). Ablation studies validate the necessity of core modules (QCA, QRL, QFC), ensuring high result credibility.
3. The paper follows top-conference structure. It clearly identifies flaws in existing methods (background), elaborates on the dual-compensation strategy (methods), and presents complete experimental data (tables), making it easy for readers to grasp the core of the work.
4. It addresses key bottlenecks in re-ID deployment (streaming inputs, edge deployment, non-query interference). PaTTA-ID provides a practical solution for adaptive re-ID in real scenarios, advancing re-ID from lab research to engineering applications.

**Weaknesses:**

1. Insufficient Technical Novelty: The core strategies are combinations of existing techniques—Input Distribution Compensation uses "query-guided sampling + contrastive learning," and Model Drift Compensation uses "camera-invariant learning + feature update." No breakthrough TTA paradigms are proposed; innovations are limited to scenario adaptation rather than core principles.
2. Incomplete Experiments: It fails to test performance on edge hardware (e.g., NVIDIA Jetson, mobile devices) (e.g., inference speed, memory usage), conflicting with its "edge deployment" positioning. It also lacks validation on complex datasets (e.g., Occluded-DukeMTMC for occlusion) and ablation of key parameters (e.g., τₛ), making it hard to prove the optimality of sampling logic.
3. Inadequate Analysis & Visualization: There is no quantitative analysis of model drift (e.g., embedding space changes over update steps) to visually justify the need for drift compensation. Critical visualizations (e.g., sample distribution before/after Input Distribution Compensation, t-SNE of cross-camera features) are missing. Streaming performance curves (performance vs. gallery accumulation steps) are also absent, inconsistent with the "streaming scenario" focus.
4. Presentation Flaws: Core module principles are vague—e.g., "adaptive adjustment of τₛ" and "sample age calculation logic" in query-guided sampling are unexplained. The number of reference format inconsistencies (e.g., missing conference/journal sources for some papers) reduces professionalism.

**Questions:**

The paper claims "edge-device deployment" but provides no performance data (inference speed, memory usage) on edge hardware (e.g., NVIDIA Jetson, mobile phones). Can you supplement edge-device performance tests to prove engineering practicality?
The threshold τₛ in query-guided sampling directly affects sample selection. The paper does not explain its value basis (e.g., why a fixed value instead of adaptive adjustment). Can you add ablation experiments on τₛ (e.g., 0.3, 0.5, 0.7) to verify parameter sensitivity?
In Model Drift Compensation, drift vectors are estimated from query feature differences between consecutive updates. How does the number of query samples (for drift vector calculation) affect estimation accuracy? Can you supplement performance comparisons under different sample sizes to clarify the basis for optimal sample size selection?
Current experiments do not test complex datasets (e.g., Occluded-DukeMTMC for occlusion, NightMarket for low light). How does the method generalize to such real-world scenarios? Can you add relevant experiments to prove robustness?
The paper claims "the dual-compensation strategy must be combined" but does not analyze the limitations of using Input Distribution Compensation or Model Drift Compensation alone. Can you add single-strategy experiments to justify the necessity of combining both? Additionally, can you provide t-SNE visualizations of cross-camera feature embeddings to intuitively show the effect of camera-invariant learning?

---

### Official Review · Reviewer_vJzK · 2025-10-31

**Soundness:** 2
**Presentation:** 3
**Contribution:** 2
**Rating:** 4
**Confidence:** 4

**Summary:**

This paper proposes a new test-time adaptation setting for person re-identification, where gallery images arrive sequentially. The method achieves strong performance and is clearly presented, but its practicality is limited by the need for predefined query sets with cross-camera annotations. Overall, the idea is interesting but needs clearer assumptions and stronger validation for real-world use.

**Strengths:**

1.	Compared with previous test-time adaptation settings that assume access to all gallery images, this work introduces a new setting where gallery images arrive sequentially. This assumption reflects more realistic scenarios in certain applications.

2.	The proposed method demonstrates a significant performance improvement over existing Re-ID and TTA approaches.

3.	Overall, the paper is clearly written and easy to follow.

**Weaknesses:**

1.	Although this work relaxes the assumption of full access to gallery images, it introduces a stronger requirement for query image annotations. Specifically, the method assumes access to a predefined query set, which requires multiple images of each person across different cameras. This assumption is not very practical for test-time adaptation, as it necessitates additional identity annotations, which are costly and time-consuming to obtain. Moreover, based on Table 4, the performance heavily depends on such data in camera-invariant learning, which limits the practical applicability of the proposed method.

2.	While the paper claims that prior works ignore the assumption that “all identities are guaranteed to exist in the gallery set,” this work still implicitly makes the same assumption. Although the gallery images are assumed to arrive sequentially, the method ultimately uses all of them and still presumes that all identities exist in the gallery. The paper would benefit from an analysis of scenarios where certain identities do not appear in the gallery.

3.	The concept of “temporal correlation” in person appearances across sequential streams is not clearly defined. Does it refer to the same person appearing at different times within the stream?

4.	Generalization to unselected queries: The model requires a predefined set of query samples, but it is unclear how well the model generalizes to unselected queries. An experimental comparison between the selected and unselected query sets would clarify this point.

5.	Number of selected queries: The model uses 20% of the total query IDs, but it is not explained how this ratio is determined. Would performance continue to improve as this ratio increases? Using a fixed number of queries instead of a dataset-dependent ratio might make the setting more generalizable across datasets and independent of data scale.

Additionally, in Figure 4, are the results evaluated only on the selected query IDs? If the evaluation includes all query IDs, the “non-adapt” results should remain consistent across settings, which does not appear to be the case.

6.	The paper does not explain how the pseudo-label mentioned in Line 216 is obtained, which causes confusion for the reader.

7.	The paper lacks an analysis of the required adaptation time, which is important for understanding the method’s efficiency and real-time applicability.

**Questions:**

1. How are the batches temporally organized? For example, are all images within a batch sampled from the same camera? This seems unlikely, as the model requires images of a person from multiple cameras. However, the paper also states that unlabeled gallery data continuously arrive from a single camera stream, which creates confusion.

2. Are adjacent batches sampled from the same or from different cameras?

---

### Note · Authors · 2025-11-13

I have read and agree with the venue's withdrawal policy on behalf of myself and my co-authors.